# Severity and Changes in OCD Dimensions during COVID-19: A Two-Year Longitudinal Study

**DOI:** 10.3390/brainsci13081151

**Published:** 2023-07-31

**Authors:** Ángel Rosa-Alcázar, Jose Luis Parada-Navas, Maria Dolores García-Hernández, Andrea Pozza, Paolo Tondi, Ana Isabel Rosa-Alcázar

**Affiliations:** 1Department of Personality, Assessment & Psychological Treatment, University of Murcia, Espinardo, 30100 Murcia, Spain; aralcazar@ucam.edu (Á.R.-A.); mariadogh@um.es (M.D.G.-H.); paolo.tondi@um.es (P.T.); 2Department of Education, University of Murcia, Espinardo, 30100 Murcia, Spain; jlpn@um.es; 3Department of Medical, Surgical and Neuroscience Sciences, University of Siena, 53100 Siena, Italy; andrea.pozza@unisi.it

**Keywords:** obsessive–compulsive disorder, OCD dimensions, COVID-19, longitudinal

## Abstract

Background: The COVID-19 pandemic appears to be associated with a worsening of obsessive-compulsive symptoms in both young people and adults with OCD and it is necessary to analyze the variables involved in this worsening over time. The main aim of this study was to examine long-term changes in total severity and obsessive-compulsive dimensions in obsessive–compulsive patients during the COVID-19 pandemic. Method: A total 250 OCD patients were selected from various associations, clinical centers and hospitals. We discarded 75 as they did not meet the inclusion criteria. A total of 175 obsessive-compulsive participants aged between 16 and 58 years old (*M* = 33.33, SD = 9.42) were evaluated in obsessive–compulsive symptom severity and dimensions OCD assessed using the Y-BOCS and D-YBOCS scales in T1 (April–June 2020) and in T2 (March–April 2022). The evaluation was carried out through an online survey and face-to-face with a professional clinician at both time points. Results: Intragroup differences in severity were observed, reaching higher scores for patients with contamination, somatic, aggressive and religious. Some patients changed their main dimension, increasing the percentage of patients with contamination and somatic obsessions. Conclusions: COVID-19 was associated with both changes in severity and also affected some dimensions more than others, particularly those related to the virus itself (contamination and somatic).

## 1. Introduction

The psychological repercussions of the COVID-19 pandemic in general and clinical populations is causing serious mental health problems, highlighting severe anxiety and depressive symptoms, severe stress levels, and posttraumatic stress [1,2,3,4].

In patients with obsessive-compulsive disorder (OCD), the elements of containment of the pandemic were associated with an increase in obsessive thinking, compulsive behavior, exacerbation of symptoms, increased suicidal ideation, internet checking, sleep disturbances, avoidance behaviors, and work difficulties [5,6,7]. Some variables are associated with more elevated worsening, such as presenting contamination symptoms, poor personal hygiene, physical distancing, avoidance of stimuli and situations, hypervigilance to somatic sensations, economic problems, reductions in social interaction, and isolation [8,9,10,11]. Some studies found significant increases in OCD contamination and cleaning symptoms during the pandemic period [12,13], while others indicated that dimensions related to hoarding, symmetry, responsibility for harm, and unacceptable thoughts have also been exacerbated, in addition to the appearance of new obsessive-compulsive symptoms [9,14,15,16]. Hygiene-related beliefs have been associated with a greater progression of symptoms from before the pandemic to its first weeks [12]. Zanjani et al. [17] evaluated the role of coping styles in the relationship between anxiety about COVID-19 and symptoms of washing-out obsessive-compulsive disorder between March and April 2020, noting that the coping strategies focused on emotion, somatization, and social support were significantly associated with washing OCD symptoms. Some variables related to an increase in the severity of obsessive responses are stress [16,18], intolerance to uncertainty [19,20], health anxiety [20], coping strategies [17], avoidance behaviors [21,22], anxiety, obsessive beliefs related to excessive responsivity, or overestimation of risk [23]. Anxiety was found to be a predictor of the development of OCD beliefs and symptoms [24]. Some authors [25] indicated that individuals under 18 years of age appeared to experience less of an impact of COVID-19 on their obsessive-compulsive symptoms than adults, explaining that this could be because they may have had less exposure to real-world triggers as they had online classes from home. Others reported that being in online treatment prevented worsening [21].

Other authors [10,22] indicated the importance of observing existing OCD cases, which may change their phenotype and the focus of their main obsessions, now related to COVID-19 contamination, specifically concerns about cleanliness and hygiene or hoarding because of fear experienced due to lack of soaps, masks, disinfectants, antiviral drugs, etc.

Although few researchers conducted longitudinal studies during the pandemic period, some compared the severity of these patients between the pre-pandemic phase and the first wave of the pandemic, showing that the presence of contamination symptoms before the first lockdown was associated with increased OCD symptom severity during the first lockdown in 2020 in a group of 30 patients with OCD [9].

In this same period, Krosnavi et al. [16] aimed to compare a group of patients with OCD before and during the pandemic (May and June 2020) both on the severity of the disorder and on the dimensions of the obsessive-compulsive symptoms, stress being a predictor factor. Results indicated higher scores in all OCD dimensions at the end, with increases in both contamination fears and other dimensions. Liao et al. [26] showed that OCD, depression, and anxiety symptoms worsened during the early stages of COVID-19, and the negative impact persisted at one-year follow-up, with optimism being a protective factor against OCD exacerbation, both during the early stages of COVID-19 and at follow-up. Rosa-Alcázar et al. [27] indicated an increase in depressive responses in schizophrenic, OCD patients and health groups, while anxiety increased only in the clinical groups, highlighting the levels of the OCD group in both variables. Benati et al. [28] evaluated the impact of the second wave of the COVID-19 pandemic through a brief cross-sectional interview, comparing results with those obtained during the first wave of the COVID-19 pandemic in the same multicenter sample of OCD outpatients. Results found more than a third of the sample worsened with increases in avoidance behaviors, suicidal ideation, and the search for security. Furthermore, men showed higher rates of occurrence of past obsessions, while women showed an increase in checking behaviors.

### Study Aims

We did not find any study that took into account the long-term effect of the pandemic or that considered OCD dimensions and the change that occurred as a result. Therefore, we aimed to evaluate the impact of the COVID-19 pandemic on a sample of patients with OCD at two time points (April–June 2020-T1 and March–April 2022, T2) assessing not only changes in OCD severity but also those experienced according to the main dimension of the patient, assessed by the DY-BOCS, in addition to analyzing whether the passage of time led some patients to change their main dimension. Specifically, the study aimed to: (1) assess severity in obsessions and compulsions according to the Y-BOCS scores and Total Y-BOCS during T1 and T2; (2) examine changes in total severity, from T1 to T2, according to the type of main obsession; (3) analyze the relationship between anxiety and depression in each obsessive dimension and if there were differences in anxiety and depression between the obsessive dimensions; and (4) assess whether patients change in the main dimensions from T1 to T2.

## 2. Material and Method

### 2.1. Participants

There were 175 participants aged between 16 and 58 years (*M* = 33.33, *SD* = 9.42) diagnosed with OCD [29] assessed at two time points (April–June 2020-T1 and March–April 2022, T2). In total, 43.4% of the sample were women. Primary obsessions during April–June 2020 were contamination (24%), aggressive (22.3%), miscellaneous (15.40%), somatic (19%), religious (10.3%), sexual (7%), and hoarding (2%). Primary compulsions were checking (40.6%), cleaning/washing (27.4%), miscellaneous (15%), repeating (12.8%), counting (1.2%), and ordering (3%). In March–April 2022, 34% patients showed new obsessions related to contamination (33.7%) and somatic (22.3%). The new compulsions were cleaning/washing (55.6%) and ordering (12.5%). The average duration of OCD was 15.25 years (SD = 9.41). In total, 36.10% patients suffered comorbidity, 67% OCD patients received pharmacological treatments (antidepressant = 69.50%, antipsychotic + antidepressant = 30.50%), and 98% were under psychological treatment. It was found that 47% were under CBT.

Inclusion criteria were: (1) a diagnosis of OCD according to DSM criteria (SCID-I, SCID-II); (2) a Yale–Brown Obsessive Compulsive Scale (Y-BOCS) [30] total score ≥ 16; (3) between 16 and 60 years of age; and (3) reaching 7 or more points in a single obsessive dimension [31]. Exclusion criteria included comorbidity with bipolar disorder, schizophrenic spectrum disorders and other psychotic disorders, personality disorders, anorexia, bulimia, disorders related to substance and addictive dependence, and neurocognitive disorders. Sample characteristics are presented in Table 1.

### 2.2. Procedure

The study met the ethical standards of the Declaration of Helsinki and was approved by the Ethics Committee of the University of Murcia (ID: 2123/2018, Spain). All participants provided written informed consent.

The recruitment process involved the following steps: (a) Contact associations/public and private clinics/hospital (April–June 2020), (b) participants engaged in an individual diagnostic interview based on DSM criteria (SCID-I, SCID-II), conducted by three clinical psychologists, (c) each OCD participant scored on the DY-BOCS dimensions. Participants scoring 7 or more points in a single dimension were included in the study. This evaluation was carried out in T1 and T2.

Responses were saved on a secured server at the University of Murcia. Participation was voluntary and free. Recruitment is shown in Figure 1.

### 2.3. Measures

-Protocol socio-demographic and clinical variables: gender, age, educational level, marital status, duration of disorder. Sociodemographic data: age, gender, civil status, education level, employment situation, changes in employment, and/or income during the pandemic.-The Yale–Brown Obsessive Compulsive Scale (Y-BOCS) [30] is comprised of 10 items assessing severity of OCD. It contains two subscales, obsessions (range = 0–20) and compulsions (range = 0–20), and a total score (range = 0–40). The scale has a high internal consistency (α = 0.87–0.90), and good convergent validity (*r* = 0.74–*r* = 0.47). A total average greater than or equal to 16 is considered of clinical significance. Cronbach’s alpha in this study was 0.87.-Dimensional Yale–Brown Obsessive Compulsive Scale [31] evaluates the presence and severity of OC symptom dimensions (aggressive, sexual, religious, symmetry, ordering, counting, contamination, hoarding, collecting, somatic, miscellaneous obsessions, and compulsions). The D-YBOCS is in two parts. The sum of these two scores corresponds to the DYBOCS total global score (ranging from 0 to 30). The scale was administered to the clinic. Cronbach’s alpha in this study was 0.85.-Hospital Anxiety and Depression Scale [32]. Self-report measure of anxiety and depression developed of 14 items rated on a 4-point Likert scale (0 a 3). It was divided into an Anxiety subscale (HADS-A) and a Depression subscale (HADS-D), both containing seven items. Cronbach’s alpha in this study was: Depression (α = 0.81), anxiety (α = 0.79) and Total (α = 0.84).

### 2.4. Data Analysis

Paired sample Student *t*-tests were performed to assess potential changes in OCD severity (obsessions, compulsions, and total Y-BOCS scores) between the scores in T1 (April–June 2020) to March–April 2022 on obsessive and compulsive symptoms (Y-BOCS).

Subsequently, a mixed ANOVA of two factors of partially repeated measures was performed on the total YBOCS variable. The inter-group factor was the type of dimension obsessive-compulsive. The intra-group factor comprised the two evaluation times (T1 and T2). The partial Eta-squared index was calculated in order to estimate the proportion of variance explained by each source.

The symmetry tests (McNemar–Bowker Test) and marginal homogeneity (Stuart–Maxwell) were performed to analyze whether the different OCD dimensions were independent of the COVID-19 situation. All participants were included in analyses. SPSS Statistic 22.00 was used for statistical analysis.

## 3. Results

### 3.1. Differences before and during COVID-19

The Student *t*-test showed a significant intragroup effect in all Y-BOCS measures (*p* < 0.001), increasing in the T2 period (see Table 2).

### 3.2. ANOVA Mixed of Group Dimensions

Table 3 presents the mean and standard deviations for the two assessment points on the total Y-BOCS. The results of the mixed ANOVAs showed a statistically significant time effect for the total Y-BOCS, with a significant increase in obsessive-compulsive symptoms across assessment points, with a medium percentage of variance accounted for (*η*^2^ ≥ 0.46). There were also significant group differences in interaction and group obsessions.

Intragroup comparisons were significant in all Y-BOCS obsessional dimensions (*p* < 0.001), except hoarding and sexual obsessions (*p* > 0.05).

The results of the intergroup comparisons were: At T1, differences were observed between the dimensions of contamination vs. hoarding (*p* > 0.001), aggressive vs. hoarding (*p* = 0.002), somatic vs. hoarding (*p* = 0.001), miscellaneous vs. hoarding (*p* = 0.021), and religious vs. hoarding (*p* = 0.053). In T2, the differences were maintained between the dimensions of contamination vs. hoarding (*p* < 0.001), aggressive vs. hoarding (*p* < 0.001), somatic vs. hoarding (*p* < 0.001), miscellaneous vs. hoarding (*p* < 0.001), religious vs. hoarding (*p* = 0.010), appearing new among the aggressive dimensions vs. hoarding (*p* < 0.001), sexual vs. somatic (*p* = 0.038), and sexual vs. contamination (*p* = 0.05). Figure 2 shows the estimated marginal means at both time points.

### 3.3. Anxiety and Depression Relationship with the Y-BOCS in Each Obsessive Dimension

Anxiety and depression did not present statistically significant differences among groups at both time points (*p* > 0.05); these variables were highly related to Total Y-BOCS in each obsessive dimension. Table 4 presents correlations between these variables.

### 3.4. Mixed ANOVA Obsessive Dimensions, Anxiety and Depression

Table 5 presents the means and standard deviations of the two assessment points for anxiety and depression. The results of the mixed ANOVAs showed a statistically significant time effect for anxiety and depression, with a percentage of variance accounted for (*η*^2^ ≥ 0.46). There were also significant group by time interactions, but no main effect of group.

### 3.5. Changes in Obsessive Dimensions at T1 and T2

Table 6 shows the frequencies and percentages of participants included in each dimension and time point. The result of the McNemar–Bowker test was statistically significant (*χ*^2^(13) = 27.44; *p* = 0.011), not fulfilling the symmetry hypothesis. The Stuart–Maxwell marginal homogeneity test reported 49 cases outside the main diagonal that collects concordances, with there being no equality of proportions in T1 and T2 (*z* = 2.834; *p* = 0.005). There was an increase in cases in contamination (24% in T1 to 33.7% in T2), with 17 participants changing to this dimension. It was likewise with the somatic dimension, which increased from 18.9% to 23.1%, with six new patients changing.

## 4. Discussion

The present study tried to answer mainly four questions: Have patients with OCD worsened from the beginning to the end of the pandemic? Has the main obsessive dimension influenced severity during this period of time? Could anxiety and depression be related to the severity of OCD within each obsessive dimension? Have patients changed their main obsessive dimension during the pandemic period?

Several studies reported worsening of obsessive-compulsive symptomatology in OCD patients during the first months of the pandemic [5,6,7]. The authors indicated some variables that might provide an explanation, such as presenting contamination symptoms, poor personal hygiene, physical distancing, avoidance of stimuli and situations, hypervigilance to somatic sensations, economic problems, reductions in social interaction, and isolation [9,10,11]. Our objective was to verify whether patients would worsen from the first wave of the pandemic until its end. Over the course of 24 months (April–June 2020-T1 until March–April 2022), we could hypothesize that patients could have worsened and, subsequently, improved, considering that the pandemic was under control and lockdown measures had been removed. Our results indicated the opposite: at the end of the pandemic, participants reached higher scores in severity, coinciding with other authors focused on the first, second wave, or a year from the onset of the pandemic [23,28]. Therefore, this expected improvement after the vaccine, decrease in mortality, elimination of isolation measures proved insufficient for the recovery of patients. Perhaps one of the most relevant variables to explain the data is stress suffered for two years with a consequent feeling of vulnerability and new stressful events, such as the war in Ukraine, economic problems, etc. [16]. In addition, it must be remembered that the pandemic affected both the mental health of the population that had pre-existing illnesses and of the general population [33].

Another objective analyzed was whether severity scores would be influenced by the main obsessive dimension of the patients. We observed that all participants worsened significantly from T1 to T2, except those whose main dimension was sexual and hoarding obsessions. However, the data must be analyzed with caution, since these dimensions included the fewest number of patients (twelve and four participants). Therefore, the increase in severity scores occurred in almost all patients, coinciding with that reported by Krosnavi et al. [16]. The present results were not in agreement with previous studies [9], where participants with contamination obsessions/compulsions were most affected, since, in the present study, the dimension scores that increased the most were somatic (4.10 points) and religious (4.17 points).

A third aim focused on analyzing the existence of differences in anxiety and depression among the obsessive dimensions. No significant differences were observed. On the contrary, the relationship between these variables and Total Y-BOCS was very high in almost all dimensions except for hoarding. However, the data must be interpreted with caution, as there were only four participants in this dimension. Correlations with depression were higher than with anxiety but both were very high at both points in time. We could, therefore, consider that chronic stress, anxiety, and depression related to COVID-19 might be underlying mechanisms that explain the exacerbation of symptoms in any of the dimensions presented. Stress due to an unpredictable external event may play an important role in both the etiology and maintenance of OCD symptoms [34]. The final aim focused on analyzing whether participants could have changed their main obsessive dimension during the pandemic period. Some authors considered that containment measures (washing, accumulation of gels and disinfectants) and fear of disease, etc., [10,22] could precipitate symptoms of contamination and washing. Our results indicated an increase in patients in the obsessive dimensions of contamination (seventeen participants) and somatic (seven participants), with the aggressive and miscellaneous dimension being where most participants were lost. This main dimension change could be related to the specific variables of the pandemic (fear of contamination, excessive washing, and fear of illness and death, etc.). Stress regarding danger of illness and contamination, the core of the COVID stress syndrome, might explain worsening of OCD patients and change of dimension in some [18]. However, many other participants remained in the same obsessive-compulsive dimension [35].

The current study has important implications for clinical practice in OCD patients. The presence of uncontrollable stressors, such as the pandemic, increases the risk that patients with OCD worsen and maintain said worsening across the very long term, with all negative consequences entailed, at personal, family, and social levels. This enables us to alleviate consequences faster, as the later we act, the greater the deterioration in this population. It must be noted that the mental consequences of the pandemic remain longer than its physical consequences. In addition, therapists should continue to provide online psychological therapy programs tailored to emerging uncontrollable situations or events, including relaxation, distress tolerance, acceptance, exposure with response prevention, and engagement in positive activities. In addition to ensuring that mental health services remain open and available during a pandemic or general crisis, care must be taken in public health messages and measures so as not to create unnecessary alarm by increasing obsessive-compulsive symptoms [24].

## 5. Limitations

This study had some limitations. Firstly, the selection of patients was not randomized. It was a cross-sectional study. The course of the epidemic together with the presence of other stressors (unemployment, family illness, financial problems, marital problems, family death, etc.) could influence changes in severity and challenge emotions. It would have been interesting to carry out more continuous follow-ups in order to verify evolution of severity of participants, in addition to comparing other types of samples, both clinical and community. Another limitation was the small number of adolescent/young participants, preventing intergenerational comparison analyses.

## 6. Conclusions

This study was the first to analyze the change at individual level during two years of the COVID-19 epidemic in Spain. Our results support the need to consider harmful effects of the pandemic on the mental health of OCD patients, since these have not disappeared even though the pandemic is controlled. Developing alternative treatments and strategies such as online consultations and digital psychiatric management during periods of chronic stress caused by uncontrollable events is a major challenge since we are facing many uncertain stressors (pandemics, wars, economic problems, etc.). The effects of the pandemic at the level of mental health in the OCD population is still relevant.

## Figures and Tables

**Figure 1 brainsci-13-01151-f001:**
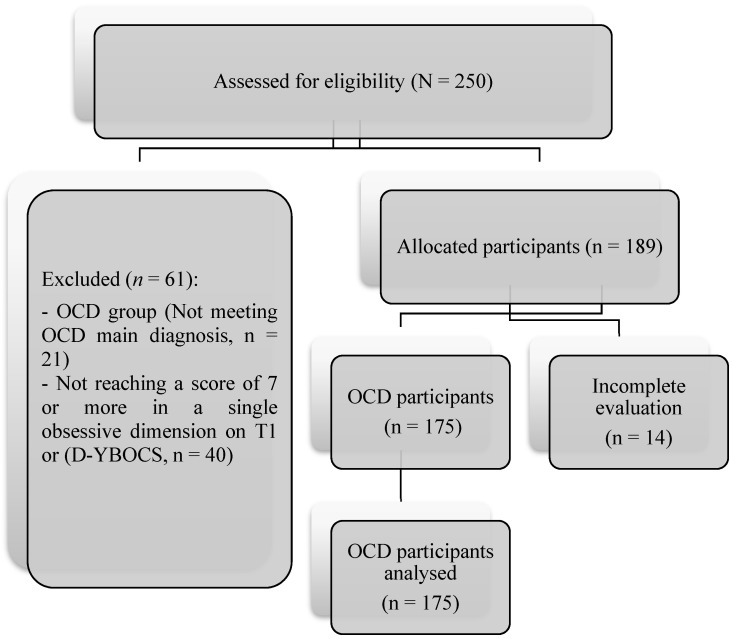
Flow diagrams of study development.

**Figure 2 brainsci-13-01151-f002:**
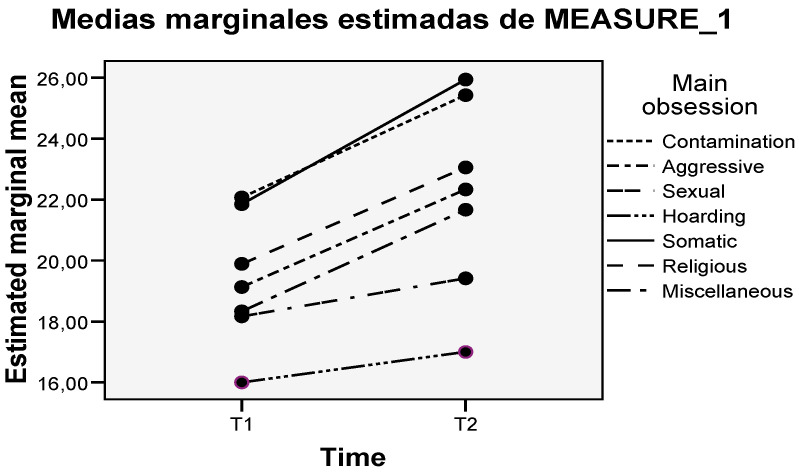
Estimated marginal means at both time points.

**Table 1 brainsci-13-01151-t001:** Sample measures.

Characteristics	OCD 2022 (*n* = 175)
Age (Mean ± SD)	33.33 ± 9.42
Sex *n* (%)	
Men	99 (56.6)
Women	76 (43.4)
Marital status *n* (%)	
Single	114 (65.1)
Married	49 (28.0)
Divorced	12 (6.9)
Educational level *n* (%)	
Elementary	9 (5.1)
Secondary education	37 (21.1)
High school	38 (21.7)
University student	91 (52.1)
Whom did you live with? (%)	
Alone	17 (9.7)
Friends/Partner/Flatmate	85 (48.6)
Family	73 (41.7)

*n* = number; SD: standard deviation.

**Table 2 brainsci-13-01151-t002:** Pairwise Student *t*-test in Y-BOCS.

	T1 (April–May 2020) Mean ± SD	T2 April–May 2022 Mean ± SD	*t*	95% I.C. Lower/Upper
Total Y-BOCS	20.16 ± 6.07	23.41 ± 7.13	−18.17 (174); *p* < 0.001	−3.59; −2.88
Y-BOCS obsessions	10.09 ± 3.06	11.77 ± 3.64	−18.18 (174); *p* < 0.001	−1.86; −1.50
Y-BOCS compulsions	10.07 ± 3.02	11.62 ± 3.53	−16.46 (174); *p* < 0.001	−1.74; −1.36

OCD = obsessive-compulsive disorder; Y-BOCS = Yale–Brown obsessive-compulsive scale. All t values were significant (*p* < 0.001).

**Table 3 brainsci-13-01151-t003:** Mixed ANOVA of group dimensions in Total Y-BOCS.

		N	T1 Mean ± SD	T2 Mean ± SD		F	*η* ^2^
Total Y-BOCS	Contamination Aggressive Sexual Hoarding Somatic Religious Miscellaneous	42 39 12 4 33 18 27	22.07 ± 7.40 19.12 ± 5.19 18.16 ± 3.95 16.00 ± 0.20 21.84 ± 6.57 19.88 ± 5.75 18.33 ± 4.55	25.42 ± 8.24 22.33 ± 6.68 19.42 ± 4.69 17.00 ± 1.15 25.93 ± 7.17 23.05 ± 7.01 21.66 ± 5.44	F (time) F (interaction) F (group)	142.86; *p* < 0.001 2.956; *p* = 0.009 2.73; *p* = 0.015	0.46 0.096 0.089

**Table 4 brainsci-13-01151-t004:** Correlation between anxiety and depression with Y-BOCS at both time points.

		Contamination	Aggressive	Sexual	Hoarding	Somatic	Religious	Miscellaneous
T1	Anxiety	0.658 **	0.626 **	0.571 *	0.098	0.488 *	0.546 *	0.384 *
	Depression	0.937 **	0.771 **	0.883 **	0.123	0.919 **	0.814 **	0.714 *
T2	Anxiety	0.710 **	0.748 **	0.633 *	0.125	0.642 **	0.689 *	0.520 *
	Depression	0.896 **	0.810 **	0.887 **	0.133	0.872 **	0.855 **	0.669 **

* *p* < 0.05; ** *p* < 0.001.

**Table 5 brainsci-13-01151-t005:** Mixed ANOVA of group dimensions in Anxiety and Depression.

		N	T1 Mean ± SD	T2 Mean ± SD		F	*η* ^2^
Anxiety	Contamination Aggressive Sexual Hoarding Somatic Religious Miscellaneous	42 39 12 4 33 18 27	10.04 ± 4.72 11.43 ± 4.89 9.16 ± 2.72 12.50 ± 0.57 13.15 ± 3.78 12.00 ± 4.18 11.93 ± 4.67	12.40 ± 4.73 13.43 ± 4.89 11.16 ± 2.72 14.75 ± 0.50 15.15 ± 3.75 14.00 ± 4.16 13.03 ± 4.59	F (time) F (interaction) F (group)	8995.25; *p* < 0.001 9.12; *p* > 0.001 2.73; *p* = 0.084	0.998 0.246 0.063
Depression	Contamination Aggressive Sexual Hoarding Somatic Religious Miscellaneous	42 39 12 4 33 18 27	8.71 ± 5.11 7.21 ± 3.64 6.58 ± 2.57 6.50 ± 0.57 9.24 ± 4.34 7.61 ± 4.43 7.22 ± 4.19	10.71 ± 5.11 9.22 ± 3.69 8.58 ± 2.56 8.50 ± 0.55 11.24 ± 4.74 9.61 ± 4.34 9.22 ± 3.50	F (time) F (interaction) F (group)	193.00; *p* < 0.001 6.12; *p* = 0.045 1.40; *p* = 0.214	0.993 0.203 0.048

**Table 6 brainsci-13-01151-t006:** Obsessional dimensions in T1 and T2.

			T2							Total
			Contamination	Aggressive	Sexual	Hoarding	Somatic	Religious	Miscella Neous	
T1	Contamination	Frequency	39	0	0	0	2	1	0	42
		%	2.3	0	0	0	1.1	0.6	0	24.0
	Aggressive	Frequency	8	24	0	0	6	1	0	39
		%	4.6	13.7	0	0	3.4	6	0	22.3
	Sexual	Frequency	0	2	9	0	1	0	0	12
		%	0	1.1	5.1	0	0.6	0	0	6.9
	Hoarding	Frequency	0	0	0	2	2	0	0	4
		%	0	0	0	1.1	1.1	0	0	2.3
	Somatic	Frequency	7	3	0	0	22	1	0	33
		%	4.0	1.7	0	0	12.6	0.6	0	18.9
	Religious	Frequency	2	1	1	0	2	12	0	18
		%	1.1	0.6	0.6	0	1.1	6.9	0	10.3
	Miscellaneous	Frequency	3	2	0	0	4	0	18	27
		%	1.7	1.1	0	0	2.3	0	10.3	15.4
Total		Frequency	59	32	10	2	39	15	18	175
		%	33.7	18.3	5.7	1.1	22.3	8.6	10.3	100

## Data Availability

The data that support the findings of this study are available on request from the corresponding author.

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
