# Peer review of "Severity and Changes in OCD Dimensions during COVID-19: A Two-Year Longitudinal Study"

_brainsci, 2023, doi:10.3390/brainsci13081151_

Round 1
Reviewer 1 Report
The paper has got a very suggestive title and a promising idea about a very relevant topic such as the severity and changes in OCD dimensions during COVID-19. I have some questions, comments and suggestions for the authors:
Abstract: there is no mention of the subject of the study in the background, only the aim of the research. "Method" is mentioned twice, I suppose it´s a mistake.
- Aims of the study are clearly presented, great job.
- Method part starts with a change in the size of the typography, please correct it. There is no description in the Method of the longitudinal study, and in my opinion is the most valuable aspect of the research. I highly recommend putting a description as 2.1
- About the sample and the inclusion criteria, they are really well described, well done. Results are wonderful and very well presented, thanks a lot. Anyway, data in text are same than data in table, so it´would be more feasible for the reader to find in the text only significative data.
Discussion.
- About the limitations, it´s necessary to explain what are the strange/modulators/mediators variables that may have influenced the results in a deeper way. Also, the differences in the sample (even intergenerational differences) are fundamental to be reflected in the paper.
- It´s really interesting to include some practical proposals in the Conclusion part. Great job again
Author Response
Dear reviewer 1,
Thank you very much for the suggested contributions to this study as this allows us to make improvements. We appreciate your invaluable time.
We have modified the areas suggested.
Faithfully,
Ana Isabel Rosa-Alcázar
Reviewer: The paper has got a very suggestive title and a promising idea about a very relevant topic such as the severity and changes in OCD dimensions during COVID-19. I have some questions, comments and suggestions for the authors:
Abstract: there is no mention of the subject of the study in the background, only the aim of the research. "Method" is mentioned twice, I suppose it´s a mistake.
Authors: We have included the following paragraph in the abstract: "The COVID-19 pandemic appears to be associated with a worsening of obsessive-compulsive symptoms in both young people and adults with OCD and it is necessary to analyse the variables involved in this worsening over time".
We have corrected the erratum on "method".
Reviewer: Aims of the study are clearly presented, great job.
Authors: We really appreciate your comment.
Reviewer: Method part starts with a change in the size of the typography, please correct it. There is no description in the Method of the longitudinal study, and in my opinion is the most valuable aspect of the research. I highly recommend putting a description as 2.1
Authors: We have corrected all typographical errors throughout the text. We have included the following sentence in participants which implies a better understanding that this is a longitudinal study. "There were 175 participants aged between 16-58 years (M = 33.33, SD= 9.42) diagnosed with OCD [30], assessed at two time points (April-June 2020- T1- and March-April 2022, T2)". In addition, the description of the study appears in section 2.2. Procedure.
Reviewer: About the sample and the inclusion criteria, they are really well described, well done.
Authors: Thank you for your comment.
Reviewer: Results are wonderful and very well presented, thanks a lot. Anyway, data in text are same than data in table, so it´would be more feasible for the reader to find in the text only significative data.
Authors: We have removed some of the data from the text.
Reviewer: Discussion. About the limitations, it´s necessary to explain what are the strange/modulators/mediators variables that may have influenced the results in a deeper way. Also, the differences in the sample (even intergenerational differences) are fundamental to be reflected in the paper.
Authors: We have included the following paragraph in the limitations section (p. 12:” This study has some limitations. Firstly, the selection of patients was not randomised. It is a cross-sectional study. The course of the epidemic together with the presence of other stressors (unemployment, family illness, financial problems, marital problems, family death, etc.) could influence changes in severity and challenge emotions. It would have been interesting to carry out more continuous follow-ups in order to verify evolution of severity of participants, in addition to comparing other types of samples, both clinical and community. Another limitation is the small number of adolescent/young participants, preventing intergenerational comparison analyses".
Reviewer: It´s really interesting to include some practical proposals in the Conclusion part. Great job again
Authors: Thank you very much again for your comments.

Reviewer 2 Report
Dear Authors,
thank you for the interesting article proposed.
The purpose of your study is to examine long-term changes in total severity and obsessive-compulsive dimensions in obsessive–compulsive patients during the COVID-19 pandemic. Despite the originality of the topic, I believe that more revisions are needed before the article can be accepted for publication in the journal.
First of all, I suggest that you report in the abstract the measurement tools used to collect the data.
I also invite you to broaden the introduction by reporting further studies that support your research hypotheses and focus on the topics you cover.
Regarding the description of the tools used for data collection, I suggest that you give examples of items and describe if there are subscales within each tool used.
I also recommend that you draw conclusions and create a section to present the possible limitations of your study.
In general, then, I suggest to review the template of because there are many oversights regarding the layout of the article (spaces, different cartteri etc.)
I hope you will accept my suggestions and that they will improve the quality of your paper.
Author Response
Dear reviewer,
First of all, thank you very much for the suggested contributions to this study as this allows us to make improvements. We appreciate your invaluable time.
We have modified the areas suggested.
Faithfully,
Ana Isabel Rosa-Alcázar
Reviewer: Dear Authors,
thank you for the interesting article proposed.
The purpose of your study is to examine long-term changes in total severity and obsessive-compulsive dimensions in obsessive–compulsive patients during the COVID-19 pandemic. Despite the originality of the topic, I believe that more revisions are needed before the article can be accepted for publication in the journal.
Authors: Thank you for your comments. We have taken your comments into consideration.
Reviewer: First of all, I suggest that you report in the abstract the measurement tools used to collect the data.
Authors: We have included in the abstract the reviewer's suggestion: "assessed using the Y-BOCS and D-YBOCS scales".
Reviewer: I also invite you to broaden the introduction by reporting further studies that support your research hypotheses and focus on the topics you cover
Authors: We have carried out the review and have included the following paragraphs in the introduction and discussion.
- 2: " anxiety, obsessive beliefs related to excessive responsivity or overestimation of risk [23]. Anxiety has been found to be a predictor of the development of OCD beliefs and symptoms [24]. Some authors [25] indicated that under-18s appeared to experience less impact of COVID-19 on their obsessive-compulsive symptoms than adults, explaining that this could be because they may have had less exposure to real-world triggers as they had online classes from home. Others reported that being in online treatment prevented worsening [26]”
- 11. "Stress due to an unpredictable external event may play an important role in both the etiology and maintenance of OCD symptoms [35]”
- 13 ". In addition to ensuring that mental health services remain open and available during a pandemic or general crisis, care must be taken in public health messages and measures so as not to create unnecessary alarm by increasing obsessive-compulsive symptoms [24]”
Reviewer Regarding the description of the tools used for data collection, I suggest that you give examples of items and describe if there are subscales within each tool used.
Authors: In the measurement section we indicate the two subscales of the Y-BOCS and D-YBOCS scales as well as the variable they measure. In the D-YBOCS scale we have included the different dimensions it measures: aggressive, sexual, religious, symmetry, ordering, counting, contamination, hoarding, collecting, somatic and miscellaneous obsessions and compulsions (p. 5)
Reviewer I also recommend that you draw conclusions and create a section to present the possible limitations of your study.
Authors: We have created a section on limitations and modified them. p. 12:” This study has some limitations. Firstly, the selection of patients was not randomised. It is a cross-sectional study. The course of the epidemic together with the presence of other stressors (unemployment, family illness, financial problems, marital problems, family death, etc.) could influence changes in severity and challenge emotions. It would have been interesting to carry out more continuous follow-ups in order to verify evolution of severity of participants, in addition to comparing other types of samples, both clinical and community. Another limitation is the small number of adolescent/young participants, preventing intergenerational comparison analyses”
Reviewer In general, then, I suggest to review the template of because there are many oversights regarding the layout of the article (spaces, different cartteri etc.)
Authors: We have checked and eliminated typographical errors throughout the manuscript.
Reviewer: I hope you will accept my suggestions and that they will improve the quality of your paper.
Authors: We have taken your suggestions into account and thus improved this manuscript. Thank you for all your advice.

Round 2
Reviewer 2 Report
Dear authors,
I’m glad you accepted my suggestions.
I think this new form of paper is clearer and more complete and I appreciate the changes you have made.
Congratulations again for your interesting study.
Author Response
Dear reviewer,
Thank you very much again. We have been able to improve the manuscript with your contributions. We appreciate your invaluable time. We thank you for your invaluable assistance.
Faithfully,
Ana Isabel Rosa-Alcazar